# Ectomycorrhizal Fungal Inoculation of *Sphaerosporella brunnea* Significantly Increased Stem Biomass of *Salix miyabeana* and Decreased Lead, Tin, and Zinc, Soil Concentrations during the Phytoremediation of an Industrial Landfill

**DOI:** 10.3390/jof6020087

**Published:** 2020-06-16

**Authors:** Dimitri J. Dagher, Frédéric E. Pitre, Mohamed Hijri

**Affiliations:** 1Institut de Recherche en Biologie Végétale, Université de Montréal and Jardin botanique de Montréal, 4101 Sherbrooke est, Montréal, QC H1X 2B2, Canada; dimitri.dagher@umontreal.ca (D.J.D.); frederic.pitre@umontreal.ca (F.E.P.); 2AgroBioSciences, Mohammed VI Polytechnic University, Lot 660, Hay Moulay Rachid, Ben Guerir 43150, Morocco

**Keywords:** arbuscular mycorrhizal fungi, ectomycorrhizal fungi, trace elements, contamination, phytoremediation, willow

## Abstract

Fast growing, high biomass willows (*Salix* sp.) have been extensively used for the phytoremediation of trace element-contaminated environments, as they have an extensive root system and they tolerate abiotic stressors such as drought and metal toxicity. Being dual mycorrhizal plants, they can engage single or simultaneous symbiotic associations with both arbuscular mycorrhizal (AM) fungi and ectomycorrhizal (EM) fungi, which can improve overall plant health and growth. The aim of this study was to test the effect of these mycorrhizal fungi on the growth and trace element (TE) extraction potential of willows. A field experiment was carried out where we grew *Salix miyabeana* clone SX67 on the site of a decommissioned industrial landfill, and inoculated the shrubs with an AM fungus *Rhizophagus irregularis*, an EM fungus *Sphaerosporella brunnea*, or a mixture of both. After two growing seasons, the willows inoculated with the EM fungus *S. brunnea* produced significantly higher biomass. Ba, Cd and Zn were found to be phytoextracted to the aerial plant biomass, where Cd presented the highest bioconcentration factor values in all treatments. Additionally, the plots where the willows received the *S. brunnea* inoculation showed a significant decrease of Cu, Pb, and Sn soil concentrations. AM fungi inoculation and dual inoculation did not significantly influence biomass production and soil TE levels.

## 1. Introduction

With the surge of the human population and urban development in the past century, modern society has been generating an ever-increasing amount of industrial waste. Industrial residues and by-products are buried in the ground around urban centers, creating vast landfills, most often composed of non-organic waste such as glass, plastics, and trace elements (TE) such as lead. With extended weathering, TE are solubilized and leech into the environment, resulting in elevated levels in soil and water [1,2]. Because they are not degradable, they accumulate in the ecosystems and pose a serious environmental and human health risk [3,4]. This makes the remediation of inorganic contaminated lands an important issue, in order to limit the spread and reduce the concentrations of these toxic elements.

Conventional methods for treating soils involve disruptive in situ and ex situ treatments such as excavation and landfill, soil washing, chemical stabilization, soil incineration, and acid leeching, among others [5]. These practices present major limitations with respect to cost, labor, and ecological footprint, as they require heavy and expensive machinery as well as complex technical procedures [6]. They also induce irreversible alterations to the soil’s physicochemical properties and microbiology [7]. Therefore, more sustainable and ecologically friendly techniques are being sought after for the remediation of TE-contaminated soils.

One such method is phytoremediation, which is the use of plants and their associated soil microbiota to degrade, extract, and stabilize pollutants [6,8,9,10]. It is cost-efficient, requires much less labor, and will ultimately help the natural revegetation of polluted sites, although it is much slower than mechanical and chemical methods. Previous experiments have demonstrated the ability of plants to assimilate and concentrate many TE such as zinc (Zn), copper (Cu), lead (Pb), cadmium (Cd), and nickel (Ni), reaching concentrations as high as 3000 mg/kg dry weight, which is a thousand-fold the normal values found in healthy plant tissue [11,12,13]. Nevertheless, results vary greatly depending on the TE and plant species/cultivar [14,15,16,17].

*Salix* is a genus comprising around 400 species of shrubs and deciduous trees that have a fast growth rate, produce high aerial biomass and deep roots, and are part of the pioneer vegetation that grows in disturbed and polluted environments [11,18,19,20,21]. Being mycorrhizal shrubs and trees, they are able to form a symbiosis with fungi that colonize their roots [22]. This association benefits the fungus with a direct access to the plant’s carbohydrates; in return, the plant takes advantage of the mycelium’s great capacity to extend beyond the root zone and absorb a greater amount of water and nutrients [23]. Moreover, arbuscular mycorrhizal (AM) fungi can improve plant health in stressful conditions such as drought [24] and trace element-polluted land [25,26].

Among mycorrhizal fungi, arbuscular (AM) and ectomycorrhizal (EM) fungi are the most commonly encountered. AM fungi are ubiquitous soil microorganisms that engage in an obligatory mutualistic association with the roots of most terrestrial plant species and play an important part in their biological functioning [23]. EM fungi, on the other hand, associate with only 10% of plant families and unlike AM fungi their mycelium does not colonize the cortical cells, but intercalates between them. These mycorrhizal associations can play an important role in the process of phytoremediation. It has been hypothesized that since they increase the plants growth, resistance to abiotic stress and survival, they can improve the plants effectiveness in extracting TEs through a higher uptake and biomass production. In addition, the AM fungi’s glycoprotein “glomalin” binds to a variety of metals and sequesters them, thus protecting the fungus and the plant from the toxic effect and immobilizing the contaminants [18,27]. Many studies have been conducted on the effects of AM fungal inoculation on phytoremediation with varying results that mostly depend on the plant/fungus combination. However, there is a paucity of information regarding the use of EM fungi in phytoremediation, and even less is known about the effect of a dual AM/EM fungal inoculation of capable trees, like *Salix* [28,29].

In this work, we investigated the combined effects of the EM (*Sphaerosporella brunnea*) and AM (*Rhizophagus irregularis*) fungi in the enhancement of willow growth, soil TE extraction and uptake by plants in a contaminated industrial brownfield in the Montreal region (Longueuil, QC, Canada). We aimed to find out whether the dual inoculation would improve or impede the willows’ aerial biomass production and TE accumulation in comparison to single or no inoculation.

## 2. Materials and Method

### 2.1. The Experimental Site

The experiment took place in an industrial landfill that was decommissioned in the 1950’s, located in the borough of Longueuil, QC, Canada (45°30′05.5″ N, 73°27′08.7″ W). The average annual temperature is 6.2 °C with 1010 mm of yearly precipitation. At the surface, the site is composed of filling material consisting largely of incineration residues, glass and metal debris. Underneath is a silty clay layer, followed by till and bedrock. Exploratory sampling of the upper layer (30 cm) found concentrations of Ba, Cd, Cu, Sn, Ni, Pb, and Zn which were above the set values by the local authorities for residential and industrial development; the “C” criterion of the Ministère de l’Environnement et la Lutte contre les changements climatiques du Québec (MELCC) (Table 1). Notably, Cu, Pb, and Zn were detected at very high concentrations, reaching 680 mg/kg, 2400 mg/kg, and 5400 mg/kg respectively (Table 1). The top layer of the landfill will be referred to from here on as “soil”. At the beginning of the project the site was totally covered by *Phragmites autralis*, which we mowed before setting up the experiment. We also removed the shoot debris and covered the ground with a black geotextile membrane to control weed growth.

### 2.2. Experimental Design and Biological Material

Five experimental blocks were set up in the field site. Each block consisted of five plots measuring 5 m by 3 m for a total block size of 15 m by 3 m. Plots were chosen randomly to receive different treatments. One plot in each block was left unplanted as a total control (CN). The remaining four plots were each planted with willow (*Salix miyabeana* clone SX67) and received one of the following treatments: (1) AM fungal inoculation (AM), (2) EM fungal inoculation (EM), (3) AM + EM fungal inoculation (XX), (4) no inoculation (SX). In June 2015, we used 20 cm cuttings that were gently hammered into the ground using a rubber mallet to a depth of approximately 10 cm. When applicable, a pilot hole was prepared in which we added 15 mL of each inoculant before inserting the cuttings. During the summer of the first year of growing season, willows were fertilized by hen manure ActiSol 5-3-2 (Notre-Dame-du-Bon-Conseil, QC, Canada), at a dose of 26 kg per 100 m^2^. This was repeated during July of the second year of growing season. Willow plantation was maintained in the two first years (2015 and 2016) where two weedings were performed to control *Phragmites autralis*. The plantation was then left without any intervention nor fertilization until 2019, when a final sampling campaign was done in November.

### 2.3. Rhizophagus irregularis

*R. irregularis* isolate DAOM-242422 (Varennes, QC, Canada) propagules were produced by cultivating AM fungal-infected transformed chicory roots on minimal medium plates [30]. After five weeks of growth, the solid medium was solubilized using citrate buffer solution [31], and the obtained mycelium/spores/infected roots mix was blended in an Eberbach blender twice for 3 s. Finally, the propagule blend was counted using a stereomicroscope and diluted in an isotonic NaCl (0.9%) solution to a concentration of 400 spores and 800 infected roots fragments per 15 mL. Field inoculation was done by pouring 15 mL of the inoculum in each hole where the willow cuttings were to be planted.

### 2.4. Sphaerosporella brunnea

A pure culture of *S. brunnea* strain Sb_GMNB300 (NRRL 66913, Perugia, Italy) [32] was kindly provided by Dr. Sánchez and was cultured in malt extract broth under constant shaking for 10 days at 24 °C. The fungal biomass was then washed with an isotonic sterile NaCl (0.9%) solution and blended in an Eberbach blender twice for two pulses of three seconds each. The propagules were then counted using a hemocytometer and diluted to 100,000 propagules per 15 mL. Field inoculation was performed by adding 15 mL of the propagule suspension in each hole where the willow clones were to be planted.

### 2.5. Sampling and Plant Measures

After two growing seasons (June 2015–October 2016), we randomly chose five plants in each of the plots for dry biomass production analysis. The selected willows were excavated from the ground and stored in individual bags for subsequent drying at 60 °C for 48 h. Due to the compacted nature of the field’s top layer it was not possible to consistently recover roots from the willows. For this reason, we only used the aerial biomass.

### 2.6. TE Concentrations

The soil was sampled in October 2016 and November 2019 from all plots for analysis of TEs. Two composite samples were prepared from each plot by combining three samples of soil for each. In 2019, five willows from each plot were randomly chosen and all stems above the last node were collected. Each treatment plot was represented by a pool of the five sampled willows for a total of 30 tissue samples (1 composite of 5 willows per plot × 4 willow treatments × 5 experimental blocks). Willow shoots were then dried in the oven at 60 °C for 72 h, after which they were ground then screened at 2 mm size. Chemical analysis was performed using a commercial service (AGAT laboratories, St-Laurent, QC, Canada). A biological concentration factor (BCF) was also calculated for each of the tested elements. BCF is the ratio of the concentration of an element in an organism to the concentration of the element in the surrounding environment [33]. In this case, it is the ratio of TE concentration in plant tissues to the TE concentration (HNO_3_ extractable) in the ground.

### 2.7. Statistical Analyses

Plant biomass, TE concentrations and BCF values were analysed using ANOVA with Tukey’s HSD post hoc comparisons with an alpha of 0.05 in JMP V.11 statistical software (SAS Institute Inc., Cary, NC, USA).

## 3. Results

### Salix myabeana Showed High Survival Rates and Cd Extraction Efficiency, but Only the EM Fungi Treatment Showed Significant Effect

After two growing seasons, all treatments showed high plant survival rates of *Salix myabeana* clone SX64, ranging from 94.4% ± 5.5 to 97.2% ± 3, with no significant statistical difference between treatments. Survival rate of *Salix* spp. clones in phytoremediation applications depends on many parameters such as pollution concentrations, climate, contamination depth, etc. For example, Guidi et al. [34] tested two *Salix* clones SX64 and SX67 to remedy a deep, polluted plume contaminated by petroleum hydrocarbons and obtained a survival rate as low as 24.4% for *S. miyabeana* clone SX67, which resulted in the failure of that phytoremediation assay [34]. The same clone SX67 of *S. miyabeana* used in our study showed an excellent survival rate, in line with previous studies of phytoremediation trials in the field [35]. Overall, we found that *S. miyabeana* clone SX67 is a suitable candidate for revegetation projects of TE-contaminated landfills, as indicated by the high survival rates observed in all treatments. This is important given that the top layer of the landfill contains little to no soil and consists of mainly rubble and debris as indicated before.

Willows inoculated with the EM fungi *S. brunnea* produced significantly higher shoot biomass than all other treatments (*p =* 0.0191) by the end of the second growing season, with an average of 72.6 g ± 8. *S. miyabeana* SX67 shoot dry weight ranged between 14 g and 153 g, with the averages for the other treatments as follows: SX, 44.4 g ± 18; XX, 53.2 g ± 18.4; AM, 53.1 g ± 7.6; EM, 72.6 g ± 8 (Figure 1). Average dry shoot biomass varied considerably between blocks (*p =* 0.0002), therefore we tested the interaction between the treatment and the block number using the “blocking” feature for ANOVA in JMP and the results were negative (no interaction), meaning the treatment effect was not different between blocks. Recently, an extensive review on dual mycorrhizal plants by Teste et al. [36] showed that the few existing studies on the subject indicate that the benefits/disadvantages of this tripartite symbiosis (including plant biomass production) vary depending on the contexts [36], including the plant and mycorrhizal species involved, as well as the life stage of the plant. On the other hand, we found that inoculation with the EM fungus *S. brunnea* significantly increased the dry stem biomass of the planted willow clones at the end of two growing seasons. EM fungi have been shown to increase plant biomass in metal-contaminated environments: Hrynkiewicz and Baum [18] found that inoculation of *Salix dasyclados* clones with the EM fungus *Amanita muscaria* grown in a metal contaminated field increased the plant stem size and biomass [18]. Similarly, Zong et al. [37] saw increased biomass of pine and oak trees grown on copper mine tailings and inoculated with a consortium of EM fungi. Moreover, a pot experiment by Ma et al. [38] showed that the inoculation of *Populus x canescens* clones with the ectomycorrhizal fungus *Paxillus involutus* increased growth and biomass production in comparison to non-inoculated plants. Our results seem to be aligned with these previous studies where EM fungi can increase plant biomass production in metal contaminated conditions.

Overall, *S. miyabeana* SX67 inoculated with mycorrhizal fungi resulted in a decreasing trend in average soil concentrations for most metals between 2016 and 2019 (Table 2). However, EM inoculation exhibited a significant decrease in the concentrations of Cu (*p* = 0.003), Sn (*p* = 0.045) and Pb (*p* = 0.046) (Figure 2 and Table 3) in comparison with other treatments. There was no significant difference between the reduction of TEs in the other treatments (SX, AM, XX) and the non-planted control (CN). As for the TE content in the plant shoots, only Ba, Cd and Zn were detected in the tissue. Their average concentrations by treatment ranged between 26.2 mg/kg ± 13.3 to 31.8 mg ± 9.7 for Ba, from 8.7 mg/kg ± 3.3 to 10.7 mg/kg ± 4.1 for Cd, and from 380 mg/kg ± 92 to 474 mg/kg ± 81 for Zn (Figure 3). There was no significant difference between treatments in the mean shoot concentrations and only the BCF of Cd was significantly higher among all treatments (*p* < 0.0001) with values between 2.11 ± 1.1 to 2.25 ± 1.8. Zinc BCF values were between 0.31 ± 0.23 and 0.54 ± 0.36 and Ba had the lowest values that ranged from 0.10 ± 0.04 to 0.15 ± 0.09 (Figure 4). BCF values did not significantly differ between treatments for all metals.

Research has shown that willows are more efficient at accumulating Cd and Zn in their aerial parts relatively to other TEs that are preferentially accumulated in the roots such as Pb and Cu [17,35,39,40,41]. Moreover, edaphic factors, such as humic/fulvic acid content, and pH, increase or decrease the TEs’ bioavailability which influences uptake by plants [42,43]. Here, Ba concentrations were within the expected range for plants (4–50 mg/kg) [44], while Cd and Zn concentrations met and even surpassed the phytotoxic ranges: 5–30 mg/kg for Cd and 100–400 mg/kg for Zn [45]. The dual inoculation did not confer any advantage in terms of shoot accumulation. However, previous studies have reported enhanced phytoextraction of TEs, induced by inoculation with either AM or EM fungi. In a pot experiment using *Populus canadensis* and *Salix viminalis* cultivars, Sell et al. [46] inoculated the soil with three EM fungi *Hebeloma crustuliniforme, Paxillus involutus* and *Pisolithus tinctorius* and found that the association of *P. tinctorius* and *P. canadensis* significantly enhanced the extraction of Cd, with concentrations in the shoots reaching 2.73 mg/kg [46]. Another pot experiment using *populus x canescens* cultivars showed that inoculation of the soil with the EM fungus *Paxillus involutus* improved the uptake and tolerance of Cd [38]. AM fungi have also been shown to increase extraction of metals such as lead as shown by Yang et al. [47] in a field experiment in Pb contaminated soil. They inoculated legume trees with the AM fungus *Rhizophagus intraradices* and found increased extraction of Pb relative to non-inoculated trees [47].

In the context of our experiment, inoculation strategies did not significantly increase TE extraction to the aerial parts by the willows. This could be in part attributed to the nature of the environment where the experiment was performed: the thin top layer of the decommissioned industrial landfill consisted of rubble and miscellaneous other materials, such as glass shards and incineration residues, making it a hostile environment that could affect the EM-plant symbioses. Among the TEs tested in this experiment, Cd showed interesting BCF values, reaching more than 3 in certain samples regardless of inoculation. Therefore, a high BCF for Cd and the increased biomass production of the EM inoculated willows suggests that the use of *S. miyabeana* SX67 inoculated with *S. brunnea* is a useful approach for the phytoextraction of Cd. Moreover, although the BCF value of zinc showed an average under 1, the actual Zn concentration in the shoots was relatively high (over 400 mg/kg), indicating that this willow cultivar can also be used for Zn contaminated environments.

In conclusion, we found that the *S. miyabeana* SX67 clone is a potential candidate for the revegetation of industrial landfills containing high concentrations of TEs, as shown by the high survival rates of the clones. Moreover, they showed an interesting potential for the phytoextraction of Cd and Zn, and possibly for the immobilization of other TEs in the roots. Fungal inoculations did not have a significant impact on the extraction of TEs, nevertheless the EM treated plants showed a significant increase in biomass production, which can lead to an increase in the total amount of extracted TEs. Whether this effect was due to the direct influence of the EM inoculation on the willows through root colonization, or the result of a change in the local microbial assemblages in reaction to the inoculation, which in turn stimulated plant growth, remains to be seen. As mentioned before, it was not possible to assess root colonization rates due to the nature of the terrain which prevented the recovery of enough fine roots. The persistence of laboratory cultivated microbial inocula in field conditions is one of the main concerns in such applications, as a host of factors modulate and influence soil and root-associated microbiota.

Future studies should be directed at finding out whether inoculation of the SX67 cultivar with mycorrhizal fungi in contaminated landfills can improve the stabilization of TEs in the roots. In addition, the fate of the indigenous soil microbes and inoculum organisms should be monitored through the evaluation of root mycorrhizal colonization, and molecular analyses of the rhizospheric microbiota using universal and specific markers as they evolve in each treatment throughout multiple growing seasons. This will shed a light on the mycorrhizal succession dynamic and its relationship with plant growth and phytoremediation performance.

## Figures and Tables

**Figure 1 jof-06-00087-f001:**
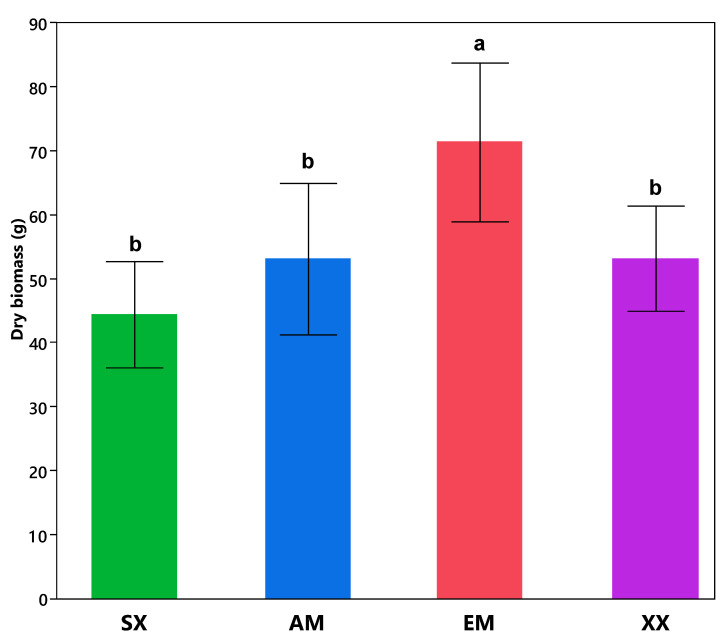
Mean dry aerial biomass production of *Salix miyabeana* SX67 inoculated with *R. irregularis* (AM), *S. brunnea* (EM), both fungi (XX), or non-inoculated (SX), illustrates that only the EM treatment significantly increased biomass. Measures were taken from five plants in each plot, at the end of the second growing season (October 2016) after drying at 60 degrees C. Column values were obtained by calculating the average of five samples per treatment in each of the five blocks, then obtaining the mean value across all blocks for each treatment. Error bars represent standard error. Treatments not sharing a letter are significantly different.

**Figure 2 jof-06-00087-f002:**
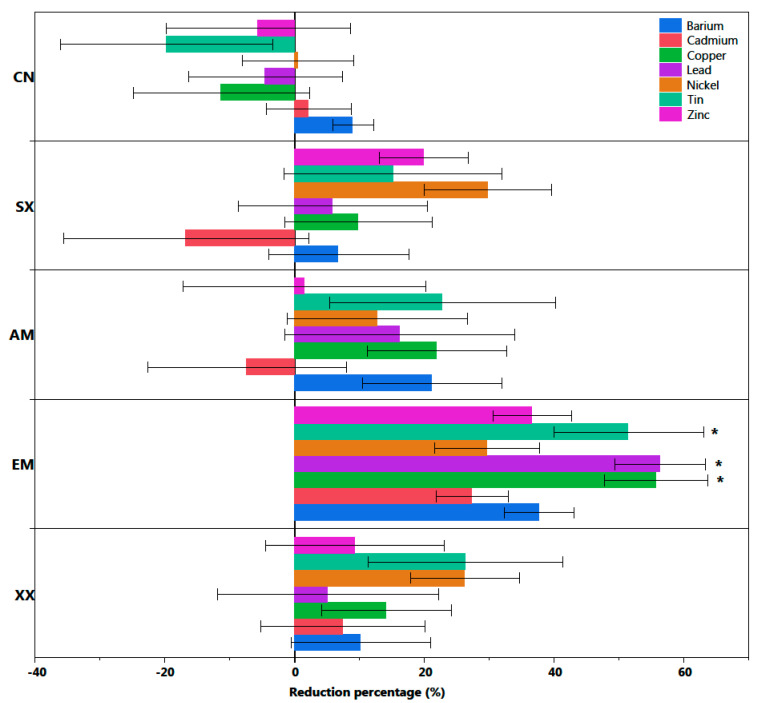
Average change in soil TE between 2016 and 2019 in five treatment plots of *Salix miyabeana* SX67 (*n* = 5 for each treatment), shows that only the EM treatment had significant decreases (Sn, Pb, and Cu). Negative values indicate an increase in concentration. Error bars represent standard error. SX = Willow non-inoculated; AM = Willow + AMF inoculation; EM = Willow + ECM inoculation; XX = Willow + AMF + ECM inoculation; CN = Non-planted control. Asterisks (*) indicate that the percent reduction of TE concentration is significantly different from other treatments

**Figure 3 jof-06-00087-f003:**
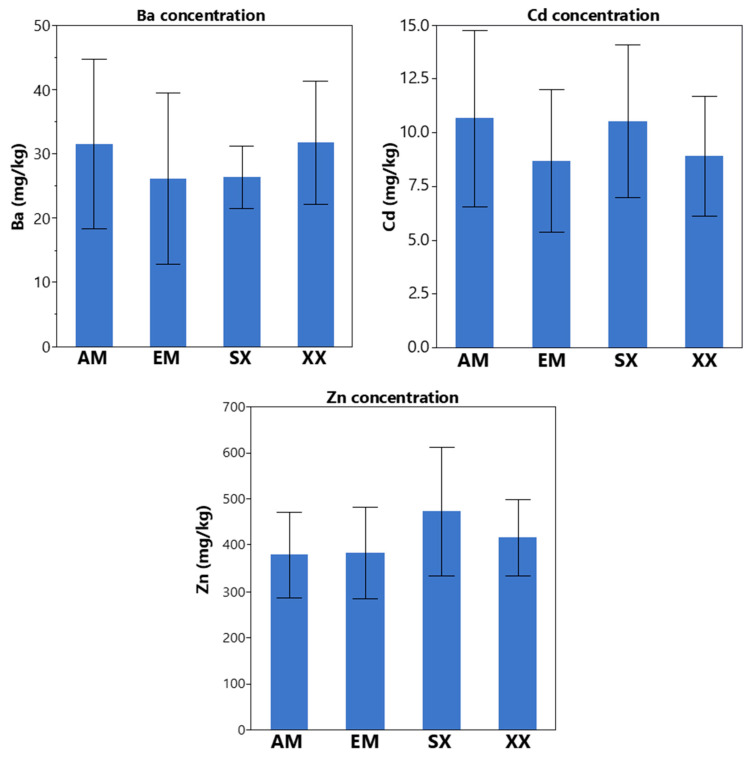
Ba, Cd, and Zn were the only TE detected in the stem tissue of the *Salix miyabeana* SX67 plots (*n* = 5 for each treatment), though there were no significant differences in the average concentrations between treatments. Error bars represent standard deviation. SX = Willow non-inoculated; AM = Willow + AMF inoculation; EM = Willow + ECM inoculation; XX = Willow + AMF + ECM inoculation; CN = Non-planted control.

**Figure 4 jof-06-00087-f004:**
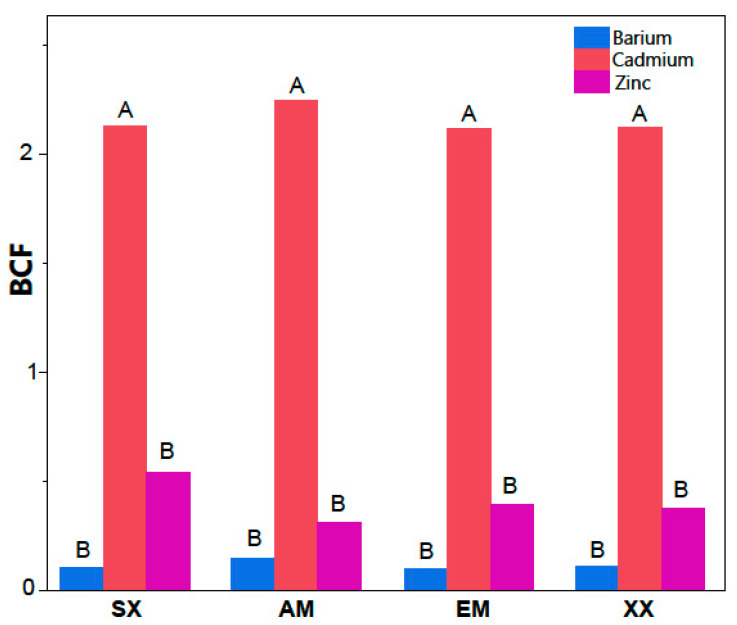
Cd had a significantly higher mean biological concentration factor (BCF) value (*p* < 0.0001) in each treatment of inoculated *Salix miyabeana* SX67 (*n* = 5) than Ba, or Zn. BCF is calculated here as the ratio of TE concentration in stem tissues to the TE concentration (HNO_3_ extractable) in the ground. Columns not sharing a letter are significantly different. AM = AMF inoculation; EM = ECM inoculation; XX = AMF + ECM inoculation; SX = Non-inoculated *S. miyabea*.

**Table 1 jof-06-00087-t001:** Trace element (TE) concentrations and pH from the 2015 exploratory soil sampling of the Longueuil, Québec industrial landfill.

		Sampling Plot				
Metal	Unit	1	2	3	4	5
Silver (Ag)	mg/kg	<0.5	8.5	3.1	0.6	2.3
Arsenic (As)	mg/kg	8	17	27	13	17
Barium (Ba)	mg/kg	190	630	530	830	540
Cadmium (Cd)	mg/kg	1.6	37	12	4	7.4
Chrome (Cr)	mg/kg	65	74	88	71	93
Cobalt (Co)	mg/kg	18	13	21	18	16
Copper(Cu)	mg/kg	110	550	680	420	600
Tin (Sn)	mg/kg	56	310	730	88	440
Manganese (Mn)	mg/kg	530	670	1200	520	790
Molybdenum (Mo)	mg/kg	2	8	12	3	7
Nickel (Ni)	mg/kg	61	65	150	88	99
Lead (Pb)	mg/kg	150	1100	2400	430	1800
Zinc (Zn)	mg/kg	380	4400	5400	2000	2300
pH		7.18	7.09	7.40	7.12	7.09

**Table 2 jof-06-00087-t002:** Mean TE concentrations and standard error of the soil per treatment as measured in 2016 and 2019 at the Longueuil, QC, industrial landfill site.

**Barium**	**CN**		**SX**		**AM**		**EM**		**XX**	
**Year**	**Mean**	**Std. Err.**	**Mean**	**Std. Err.**	**Mean**	**Std. Err.**	**Mean**	**Std. Err.**	**Mean**	**Std. Err.**
**2016**	262.7	38.33	337.5	47.94	339.8	52.62	464.8	64.19	362.9	39.5
**2019**	247.9	35.27	298.1	38.43	247.5	39.8	264.3 *	22.84	302.3	38.85
**Cadmium**	**CN**		**SX**		**AM**		**EM**		**XX**	
**Year**	**Mean**	**Std. Err.**	**Mean**	**Std. Err.**	**Mean**	**Std. Err.**	**Mean**	**Std. Err.**	**Mean**	**Std. Err.**
**2016**	5.64	1.03	7.6	1.91	7.16	1.28	6.99	1.02	6.35	1.23
**2019**	5.4	1.08	7.17	1.71	7.55	1.58	4.83 *	0.72	4.89	0.75
**Copper**	**CN**		**SX**		**AM**		**EM**		**XX**	
**Year**	**Mean**	**Std. Err.**	**Mean**	**Std. Err.**	**Mean**	**Std. Err.**	**Mean**	**Std. Err.**	**Mean**	**Std. Err.**
**2016**	375.4	74.65	377.8	88.09	948.8	295.88	1261.6	476.76	526.7	94.17
**2019**	601.4	265.34	324.2	68.66	581.6	105.64	351.4	65.09	390.3 *	61.04
**Lead**	**CN**		**SX**		**AM**		**EM**		**XX**	
**Year**	**Mean**	**Std. Err.**	**Mean**	**Std. Err.**	**Mean**	**Std. Err.**	**Mean**	**Std. Err.**	**Mean**	**Std. Err.**
**2016**	587.2	128.68	781.5	208.51	1010.4	178.19	1334	258.36	1672.7	931.39
**2019**	557	123.23	713.7	194.74	792	175.93	594.3 *	156.94	622.7	124.49
**Nickel**	**CN**		**SX**		**AM**		**EM**		**XX**	
**Year**	**Mean**	**Std. Err.**	**Mean**	**Std. Err.**	**Mean**	**Std. Err.**	**Mean**	**Std. Err.**	**Mean**	**Std. Err.**
**2016**	87.1	9.29	107.2	12.74	114.1	13.45	118.2	12.89	93.2	11.67
**2019**	88.6	13.04	68.5 *	6.99	95.9	16.47	79.5 *	9.98	62.6 *	4.42
**Tin**	**CN**		**SX**		**AM**		**EM**		**XX**	
**Year**	**Mean**	**Std. Err.**	**Mean**	**Std. Err.**	**Mean**	**Std. Err.**	**Mean**	**Std. Err.**	**Mean**	**Std. Err.**
**2016**	225.5	60.46	337.6	111.24	559.6	117.66	542.8	101.18	497.5	155.08
**2019**	244.6	62.51	247	72.06	410.7	123.79	265.9 *	88.58	218.5	53.74
**Zinc**	**CN**		**SX**		**AM**		**EM**		**XX**	
**Year**	**Mean**	**Std. Err.**	**Mean**	**Std. Err.**	**Mean**	**Std. Err.**	**Mean**	**Std. Err.**	**Mean**	**Std. Err.**
**2016**	1753.9	300.35	1697.7	324.05	1992.7	325.54	2063.6	249.48	2001.3	355.89
**2019**	2070.1	393.19	1346 *	280.25	1656.3	259.44	1363.6 *	236.71	1455.2	236.43

*n* = 5 for each year and treatment combination. For each treatment, asterisks (*) show that the mean TE concentration was significantly different from the previous measurement (2016 vs. 2019). CN = Non-planted control; SX = Willow non-inoculated; AM = Willow + AMF inoculation; EM = Willow + ECM inoculation; XX = Willow + AMF + ECM inoculation; Std. Err. = Standard error.

**Table 3 jof-06-00087-t003:** ANOVA analysis of the aerial dry biomass of *S. miyabeana* SX67 and the TE percentage decrease/increase in the soil for each treatment after two growing seasons at the Longueuil, Québec industrial landfill site.

	Metal Decrease/Increase % and Plant Dry Biomass Tukey’s HSD Comparisons				
	**CN**	**SX**	**AM**	**EM**	**XX**
***Ba***	N.S.	N.S.	N.S.	N.S.	N.S.
***Cd***	N.S.	N.S.	N.S.	N.S.	N.S.
***Cu***	B	B	AB	A	AB
***Pb***	B	AB	AB	A	AB
***Ni***	N.S.	N.S.	N.S.	N.S.	N.S.
***Sn***	B	AB	AB	A	AB
***Zn***	N.S.	N.S.	N.S.	N.S.	N.S.
***Biomass***		B	B	A	B

Within each row, treatments not sharing a letter are significantly (*p* < 0.05) different. N.S. = Not significant. CN = Non-planted control; SX = Willow non-inoculated; AM = Willow + AMF inoculation; EM = Willow + ECM inoculation; XX = Willow + AMF + ECM inoculation.

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
