# Peer review of "Ectomycorrhizal Fungal Inoculation of Sphaerosporella brunnea Significantly Increased Stem Biomass of Salix miyabeana and Decreased Lead, Tin, and Zinc, Soil Concentrations during the Phytoremediation of an Industrial Landfill"

_jof, 2020, doi:10.3390/jof6020087_

Round 1

Reviewer 1 Report

This study investigate the effectiveness of phytoremediation via mycorrhization of willow trees in contaminated soil. Using a time-consuming woody plant materials, this study focuses on the inoculation effects of exotic EMF and AMF, with concise objectives and precise experimental design.
The finding of a situation-dependent inoculation effect is not new, but includes some noteworthy findings, such as the absence of an additive effect of EMF and AMF.

I think the claim is worthy of acceptance as it stands, but I'd like to make a few comments.

1) I think the abstract should describe the results of the AMF treatment.

2) line 98: The authors should remove one removed.

3) The last part of the discussion: The presence or absence of the inoculation effect of exotic mycorrhizal fungi is greatly influenced by its interaction with indigenous mycorrhizal fungi. The future prospects for the analysis of indigenous fungi should also be discussed.

Author Response

Reviewer#1

This study investigate the effectiveness of phytoremediation via mycorrhization of willow trees in contaminated soil. Using a time-consuming woody plant materials, this study focuses on the inoculation effects of exotic EMF and AMF, with concise objectives and precise experimental design.
The finding of a situation-dependent inoculation effect is not new, but includes some noteworthy findings, such as the absence of an additive effect of EMF and AMF.

I think the claim is worthy of acceptance as it stands, but I'd like to make a few comments.

1) I think the abstract should describe the results of the AMF treatment.

  • We added “AM fungi inoculation and dual inoculation did not significantly influence biomass production and soil TE levels.” At the end of the abstract

2) line 98: The authors should remove one removed.

  • Done

3) The last part of the discussion: The presence or absence of the inoculation effect of exotic mycorrhizal fungi is greatly influenced by its interaction with indigenous mycorrhizal fungi. The future prospects for the analysis of indigenous fungi should also be discussed.

  • We thank the reviewer for this interesting suggestion which was considered in the discussion (Page 6, L 251-264) as follow “Whether this effect was due to the direct influence of the EM inoculation on the willows through root colonization, or the result of a change in the local microbial assemblages in reaction to the inoculation, which in turn stimulated plant growth remains to be seen. As mentioned before, it was not possible to assess root colonization rates due to the nature of the terrain which prevented the recovery of enough fine roots. The persistence of laboratory cultivated microbial inocula in field conditions is one of the main concerns in such applications, as a host of factors modulate and influence soil and root-associated microbiota.
  • Future studies should be directed at finding out whether inoculation of the SX67 cultivar with mycorrhizal fungi in contaminated landfills can improve the stabilization of TEs in the roots. In addition, the fate of the indigenous soil microbes and inoculum organisms in the rhizosphere should be monitored through the evaluation of root mycorrhizal colonization, and molecular analyses of the rhizospheric microbiota using universal and specific markers, as they evolve in each treatment throughout multiple growing seasons. This will shed a light on the mycorrhizal succession dynamic and its relationship with plant growth and phytoremediation performance.”

Reviewer 2 Report

This is a nice, mostly descriptive study. The authors tested the potential of mycorrhizal inoculation to ameliorate phytoremediation practices using willow. The study is sound and experiments well designed. It is however a pity that the authors did not evaluate success of inoculation; assess mycorrhizal colonization during or at the end of the experiment. Did the fungi colonize the willow trees after all or are the observed effects rather due to secondary effects (e.g. changes in microbial community due to the introduction of inoculum)? Could the authors comment on this and frame the results, conclusions to include this?

minor comments:

ln65 arbuscular (EM); change EM into AM

ln141 please rephrase. Sentence is not clear.

Table 3. Please explain abbreviations CN/SX/AM/EM/XX as in the figures.

Author Response

Reviewer#2

This is a nice, mostly descriptive study. The authors tested the potential of mycorrhizal inoculation to ameliorate phytoremediation practices using willow. The study is sound and experiments well designed. It is however a pity that the authors did not evaluate success of inoculation; assess mycorrhizal colonization during or at the end of the experiment. Did the fungi colonize the willow trees after all or are the observed effects rather due to secondary effects (e.g. changes in microbial community due to the introduction of inoculum)? Could the authors comment on this and frame the results, conclusions to include this?

  • We agree with the reviewer’s comment that the mycorrhizal colonization of root with EM and AM fingi would have been informative of the success of our inoculation. To mitigate our claim, we have added at the discussion section as follows ”Whether this effect was due to the direct influence of the EM inoculation on the willows, or the result of a change in the local microbial assemblages in reaction to the inoculation, which in turn stimulated plant growth remains to be seen. As mentioned before, it was not possible to assess root colonization rates due to the nature of the terrain which prevented the recovery of enough fine roots.”

minor comments:

ln65 arbuscular (EM); change EM into AM

  • Done

ln141 please rephrase. Sentence is not clear.

  • The sentence now reads as follows: “In 2019, five willows from each plot and were randomly chosen and all stems above the last node were collected”

Table 3. Please explain abbreviations CN/SX/AM/EM/XX as in the figures.

  • We added the following to the legend: CN = Non-planted control; SX = Willow non-inoculated; AM = Willow + AMF inoculation; EM = Willow + ECM inoculation; XX = Willow + AMF + ECM inoculation

Reviewer 3 Report

This is a well written paper based on a nicely designed experiment, which tested the inoculation of AM and EM fungi, and their mixture,  on the growth of willow and the TEs uptakes. The results are interesting which should certainly be published. The paper is in a good shape, and I only have several small annotations as attached. The title is a bit long and boring, which could  be shortened. 

Author Response

Reviewer#3

This is a well written paper based on a nicely designed experiment, which tested the inoculation of AM and EM fungi, and their mixture,  on the growth of willow and the TEs uptakes. The results are interesting which should certainly be published. The paper is in a good shape, and I only have several small annotations as attached.

The title is a bit long and boring, which could  be shortened. 

  • We thank the reviewer for this suggestion. We have now shortenen the title by removing “trace elements-contminated”.
  • Line 65: We replaced AM by EM as suggested
  • Line 229: we replaced “Moreover, while Zn showed BCF values averaging under 1” with “Although the BCF value of zinc showed an average under 1”
  • Table 2: We added the mention of the asterisks
  • Figure 2: We added “Asterisks (*) indicate that the percent reduction of TE concentration is significantly different from other treatments”